# Drug use for gastrointestinal symptoms during pregnancy: A French nationwide study 2010–2018

Antoine Meyer [1,2,3]*, Marion Fermaut[4], Jérôme Drouin[1], Franck Carbonnel[2,3], Alain Weill[1]

**1** GIS-EPIPHARE, Épidémiologie des produits de santé, ANSM-CNAM, 42 bd de la Libération, Saint Denis, France, **2** Assistance Publique-Hôpitaux de Paris, Hôpital Bicêtre, Le Kremlin Bicêtre, France, **3** Université Paris Sud, Le Kremlin Bicêtre, France, **4** Assistance Publique-Hôpitaux de Paris, Hôpital Jean Verdier, Bondy, France

* antoinemeyer@gmail.com

## Abstract

### Purpose

To describe drug prescription for gastrointestinal symptoms during pregnancy.

### Methods

Using the French national health database, we identified pregnancies ending with a birth between April 2010 and December 2018, in France. We studied prescription of antacids, antispasmodics, antinauseants, laxatives and antidiarrheals during pregnancy, between two trimesters before and two trimesters after delivery. We also assessed hospitalization for gastrointestinal symptoms during pregnancy.

### Results

Among 6,365,471 pregnancies, 4,452,779 (74.0%) received at least one gastrointestinal drug during pregnancy; 2,228,275 (37.0%) received an antacid, 3,096,858 (51.5%) an antispasmodic, 1,861,731 (31.0%) an antinauseant, 919,116 (15.3%) a laxative and 617,808 (10.3%) an antidiarrheal. Prescription of proton pump inhibitors doubled from 12.2% in 2010 to 26.0% in 2018, while domperidone use decreased from 18.3% in 2010 to 2.2% in 2018. In addition, prescription of antacids increased from 7.0% during the trimester before pregnancy to 11.8% during the 1st trimester, 17.0% during the 2nd trimester and 23.4% during the 3rd trimester. Antispasmodic use was 10.6% during the trimester before pregnancy, 23.1% during the 1st trimester, 25.2% during the 2nd trimester and 24.0% during the 3rd trimester. Prescription of antinauseant drugs increased from 5.0% during the trimester before pregnancy to 25.7% during the 1st trimester, then decreased to 6.4% during the 2nd trimester and 3.2% during the 3rd trimester. Nausea/vomiting was the most common cause of hospitalization for gastrointestinal symptoms or diseases during pregnancy, although it accounted for only 1.0% of pregnancies.

**Data Availability Statement:** The authors had access to the SNDS database in application of the provisions of Articles R. 1461-12 et seq. of the French Public Health Code and the French data protection authority decision CNIL-2016-316.

Future researchers can request access via the Health data hub: (https://documentation-snds. health-data-hub.fr/introduction/03-acces-snds. html#les-acces-sur-projet).

**Funding:** The author(s) received no specific funding for this work.

**Competing interests:** Conflict of Interest: - Antoine Meyer: no conflict of interest. - Marion Fermaut: no conflict of interest. - Jérôme Drouin: no conflict of interest. - Franck Carbonnel: received honoraria from Amgen, BMS, Celltrion, Enterome, Ferring, Janssen, Medtronic, Pfizer, Pharmacosmos, Roche and Tillotts as well as lecture fees from Abbvie, Astra, BMS, Ferring, Janssen, MSD, Pfizer, Pileje, and Takeda. - Alain Weill: no conflict of interest. This does not alter our adherence to PLOS ONE policies on sharing data and materials.

## Conclusions

Approximately three-quarters of women use drugs for gastrointestinal symptoms during pregnancy in France. Prescription of gastrointestinal drugs during pregnancy should be the subject of more detailed risk-benefit assessment and recommendations.

## Background and aims

Thalidomide was used as an antinauseant medication during pregnancy until its teratogenic effect was demonstrated in the 1960s. This scandal led to an increased awareness of the safe use of drugs during pregnancy [1]. Yet, data on the safety of drugs used during pregnancy (for women and offspring) is often lacking as pre-marketing clinical trials exclude pregnant women. Observational studies are therefore essential to study the efficacy and safety drugs in pregnant women and their offspring. For instance, a French nationwide study on recent data confirmed that valproic acid, a treatment for epilepsy, cause birth defects and delayed cognitive development [2, 3].

Despite the concerns about the safety of medication use during pregnancy, many women take medication during pregnancy, but with considerable variability according to geographical region and over time [4, 5]. In recent years, between 58% and 97% of women take at least one medication during pregnancy, depending on the country [4–7]. Gastrointestinal symptoms are common during pregnancy: 50 to 90% of pregnant women experience nausea and 30–80% experience heartburn [8, 9]. No study has specifically assessed the use of the various classes of gastrointestinal drugs during pregnancy, as these drugs are often grouped into a single category of gastrointestinal drugs [4, 6, 7, 10, 11]. Drug use studies in pregnancy are useful to identify the most frequently used drugs and thereby decide which drug safety studies should be prioritized.

The aim of the present study was to describe drug use for gastrointestinal symptoms during pregnancy in a large-scale, population-based, nationwide study between 2010 and 2018 using the French national health data system, in a partnership between clinicians, French national health insurance (*Caisse Nationale d'Assurance Maladie*) and the French National Agency for Medicines and Health Products Safety (*Agence Nationale de Sécurité du Médicament*).

## Materials and methods

### Data source

The French national health data system (*Système National des Données de Santé*, SNDS) covers more than 99% of the French population (around 66,000,000 people). Each person is identified by a unique, anonymous number. The SNDS contains all outpatient information (demographics, drugs dispensed, and procedures) and all inpatient information (expensive drugs dispensed, procedures performed during hospital stays, and diagnoses) [12]. Studies based on this database have produced meaningful results in recent years [13–17].

We adapted an algorithm previously developed and used [16, 18] to identify pregnancies ending between April 1, 2010 and December 31, 2018 among women aged 15 to 49 in the SNDS (S1 Table). The pregnancy end date was the date of delivery, or, when missing, the date of admission for pregnancy completion. The pregnancy start date was calculated using gestational age at the end of pregnancy, or, when missing, the date of the last menstrual period entered by the physician at the end of pregnancy [19].

## Study population

All pregnancies ending with a birth—either livebirth—between April 2010 and December 2018 were included. A stillbirth was the delivery of a dead fetus after 22 weeks of amenorrhea (referred to as weeks thereafter). We excluded pregnancies with elective or therapeutic abortions, spontaneous abortions, ectopic pregnancies and hydatidiform moles and other abnormal products of conception (blighted ovum and nonhydatidiform moles), because to study drug exposure throughout pregnancy, pregnancies that last three trimesters are needed. Pregnant women with no healthcare record in the SNDS database during the 2 years preceding the pregnancy were excluded. These women may have lived outside of France or their outpatient data may have been unavailable in the SNDS (less than 1% of the French population).

## Drug exposure and hospitalizations

We studied each drug dispensed for gastrointestinal symptoms including: antacids, antispasmodics, antinauseants, laxatives, antidiarrheals and other drugs for functional gastrointestinal disorders. Details on International Non-Proprietary name and Anatomical Therapeutic Chemical Classification are given in S2 Table. For each pregnancy, seven trimesters were studied: two trimesters before the beginning of pregnancy (Trim-2: day −182 to day −92; Trim-1: day −91 to day −1), each trimester of pregnancy (Trim1: day 0 i.e. fertilisation to day 90; Trim2: day 91 to day 181; Trim3: day 182 to delivery −1) and two trimesters after the end of pregnancy (Trim+1: delivery to delivery + 91; Trim+2: delivery + 92 to delivery + 182). A pregnancy trimester was considered to be exposed to a drug when this drug was dispensed at least once during this trimester. We also grouped these drugs into a single category to study the rate of pregnant women using at least one gastrointestinal drug. Additionally, we looked at the dispensing rates of drugs in the Anatomical Therapeutic Chemical (ATC) classes A02 to A09 to allow more reliable comparison with published data.

We also studied the most common reasons for hospitalization for a gastrointestinal symptom or disease between two trimesters before and two trimesters after pregnancy, including: nausea/vomiting, cholestasis, proctological disease (fissure, fistula, abscess, hemorrhoids), appendicitis and biliary diseases (S3 Table).

## Maternal and pregnancy characteristics

Baseline maternal characteristics included: age, Complementary Universal Health Insurance status (free access to health care for people with low income), a deprivation index expressed in quintiles that was developed in France as the first component of a principal component analysis of 4 socioeconomic variables [20], and income (general health insurance scheme: none, <€2,000/month, ≥€2,000/month; and agricultural/self-employed scheme) calculated from the woman's salary during the three months before maternity leave. Gravidity (1st pregnancy, 2nd pregnancy,. . .) was defined as the pregnancy number in a woman during the study period. Assisted reproduction was defined as any of the following procedures performed in France during the two months preceding pregnancy: ovarian stimulation, oocyte retrieval, artificial insemination or *in vitro* fertilization. The clinical setting of termination of pregnancy was recorded: university hospital, general hospital, private hospital, or outpatient procedure.

Pregnancy characteristics included type of delivery (cesarean or vaginal), vital status at birth (livebirth or stillbirth), prematurity (births occurring before 37 weeks were considered to be preterm and those occurring before 32 weeks were considered to be very preterm) and birth weight for gestational age (below the 10th percentile and above the 90th percentile of the gestational age computed in the national pregnancy cohort were considered small and large

for gestational age, respectively). Birth weight was available for those infants in whom linkage between mother and child data was available (78.5% of pregnancies with delivery).

## Statistical analysis

The unit of analysis was a pregnancy, i.e. all of a patient's pregnancies were included in the analysis. We first described maternal characteristics at the beginning of pregnancy and pregnancy characteristics for each pregnancy: median and interquartile range (IQR) for continuous variables and proportions for categorical variables. Medication dispensing during pregnancy over time was then described for pregnancy ending between April 2010 and December 2018: crude numbers and percentages of exposed pregnancies by year of pregnancy end. Medication dispensing and hospitalizations before, during and after pregnancy was described for pregnancies ending between April 2010 and June 2018: crude numbers and percentages of exposed pregnancies by trimester. A sensitivity analysis was performed and excluded pregnancies of less than 37 weeks.

All analyses were performed with SAS® software version 9.4 (SAS Institute, North Carolina, USA). The French public institution which conducted this study has permanent access to the SNDS database in application of the provisions of Articles R. 1461–12 *et seq*. of the French Public Health Code and the French data protection authority decision CNIL-2016-316. No informed consent was therefore required. This research did not receive any funding.

## Results

### Study population

Among approximately 32 million women in France, 4,546,505 completed a pregnancy between April 1, 2010 and December 31, 2018. A total of 8,796,626 pregnancies were identified among these women. The following pregnancies were excluded: 1,687,516 elective/therapeutic abortions, 339,553 spontaneous abortions, 96,073 ectopic pregnancies, 110,902 hydatidiform moles or other abnormal products of conception and 197,111 with no prior outpatient health care utilization during the 2 years before pregnancy. A total of 6,365,471 pregnancies ending with delivery were therefore included (S1 Fig). The annual number of pregnancies decreased slightly from 763,069 in 2011 to 682,065 in 2018 (S2 Fig).

### Maternal and pregnancy characteristics

Maternal and pregnancy characteristics are presented in Table 1. Median age at the beginning of pregnancy was 29 years (IQR: 26–33), 809,034 (12.7%) women had Complementary Universal Health Insurance cover, 2,508,345 (39.4%) had no income and 174,174 (2.7%) had undergone assisted reproduction. Pregnancies ended with 1,273,816 (20.0%) cesarean sections, 32,677 (0.5%) stillbirths, 352,817 (5.5%) preterm births, 73,135 (1.2%) very preterm births and the mean birth weight was 3.29 kg (IQR: 2.98–3.61).

### Drug exposure

Among all pregnancies with delivery, 4,452,779 (74.0%) received at least one gastrointestinal drug during pregnancy; 2,228,275 (37.0%) pregnancies were exposed to an antacid, 3,096,858 (51.5%) to an antispasmodic, 1,861,731 (31.0%) to an antinauseant, 919,116 (15.3%) to a laxative and 617,808 (10.3%) to an antidiarrheal (Table 2). The antispasmodic phloroglucinol was by far the most commonly prescribed drug (51.0% of all pregnancies). These rates of drug exposure during pregnancy were consistent over ages at the beginning of pregnancy and excluding pregnancies of less than 37 weeks (S3 Fig and S4 Table). However, drug use was

**Table 1. Demographic data and pregnancy characteristics (in thousands).**

| | Total | Age at the beginning of pregnancy (years) | | |
| --- | --- | --- | --- | --- |
| | | 15–24 | 25–34 | 35–49 |
| | N, thousands (%) | N, thousands (%) | N, thousands (%) | N, thousands (%) |
| **Number of pregnancies** | 6 365 | 1 155 | 4 108 | 1 103 |
| **Age (years)¶** | 29.0 [26.0–33.0] | 22.0 [20.0–24.0] | 29.0 [27.0–32.0] | 37.0 [36.0–39.0] |
| **Income** | | | | |
| General health scheme: No income | 2 508 (39.4) | 571 (49.5) | 1 479 (36.0) | 458 (41.6) |
| General health scheme: < €2,000/month | 1 856 (29.2) | 441 (38.2) | 1 181 (28.8) | 234 (21.2) |
| General health scheme: ≥ €2,000/month | 1 603 (25.2) | 89 (7.7) | 1 185 (28.8) | 329 (29.9) |
| Agricultural and self-employed scheme | 398 (6.3) | 54 (4.7) | 263 (6.4) | 81 (7.4) |
| **Deprivation index** | | | | |
| Quintile 1 (less deprived) | 1 204 (19.8) | 114 (10.7) | 816 (20.6) | 274 (26.0) |
| Quintile 2 | 1 208 (19.9) | 174 (16.3) | 820 (20.7) | 214 (20.4) |
| Quintile 3 | 1 191 (19.6) | 210 (19.7) | 786 (19.8) | 195 (18.6) |
| Quintile 4 | 1 179 (19.4) | 243 (22.8) | 759 (19.2) | 178 (16.9) |
| Quintile 5 (more deprived) | 1 297 (21.3) | 326 (30.5) | 782 (19.7) | 190 (18.1) |
| **Complementary Universal Health Insurance§** | 809 (12.7) | 281 (24.4) | 403 (9.8) | 125 (11.4) |
| **Gravidity (Apr 1, 2010—Dec 31, 2018)** | | | | |
| 1st pregnancy | 3 863 (60.7) | 798 (69.1) | 2 431 (59.2) | 634 (57.5) |
| 2nd pregnancy | 1 858 (29.2) | 268 (23.3) | 1 256 (30.6) | 334 (30.3) |
| 3rd and more pregnancy | 645 (10.1) | 88 (7.7) | 421 (10.3) | 135 (12.3) |
| **Assisted reproduction** | 174 (2.7) | 6 (0.5) | 112 (2.7) | 56 (5.1) |
| **Place of end of pregnancy** | | | | |
| University hospital | 1 172 (18.4) | 209 (18.1) | 733 (17.8) | 230 (20.9) |
| General hospital | 3 529 (55.5) | 698 (60.5) | 2 261 (55.1) | 570 (51.7) |
| Private hospital | 1 656 (26.0) | 248 (21.4) | 1 108 (27.0) | 301 (27.3) |
| Outpatient | 9 (0.1) | 1 (0.1) | 6 (0.2) | 2 (0.2) |
| **Multiple pregnancy** | 62 (1.3) | 7 (0.8) | 40 (1.3) | 15 (1.7) |
| **Cesarean section** | 1 274 (20.0) | 184 (15.9) | 788 (19.2) | 302 (27.4) |
| **Pregnancy outcome** | | | | |
| Livebirth | 6 333 (99.5) | 1 148 (99.4) | 4 089 (99.5) | 1 095 (99.3) |
| Stillbirth | 33 (0.5) | 7 (0.6) | 19 (0.5) | 7 (0.7) |
| **Pregnancy term** | | | | |
| Term birth (≥37 weeks) | 5 940 (93.3) | 1 071 (92.7) | 3 852 (93.8) | 1 017 (92.2) |
| Preterm birth (<37 weeks) | 353 (5.5) | 68 (5.9) | 214 (5.2) | 70 (6.4) |
| Very preterm birth (<32 weeks) | 73 (1.2) | 15 (1.3) | 42 (1.0) | 16 (1.4) |
| **Birth weight (kg)¶** | 3.29 [2.98–3.61] | 3.24 [2.93–3.55] | 3.30 [3.00–3.62] | 3.30 [2.97–3.63] |
| **Birth weight for gestational age** | | | | |
| Small for gestational age (<P10) | 492 (9.8) | 106 (11.7) | 302 (9.4) | 84 (9.7) |
| Appropriate (P10-P90) | 4 010 (80.2) | 731 (80.5) | 2 600 (80.6) | 680 (78.7) |
| Large for gestational age (>P90) | 497 (10.0) | 71 (7.8) | 326 (10.1) | 100 (11.6) |

¶Quantitative variable, median [interquartile range].

§Free access to health care for people with low income. P10/P90: 10th/90th percentile.

slightly higher in more deprived pregnancies than in less deprived pregnancies (76.0% vs 72.3% for at least one gastrointestinal drug use), and this applied to most of the drugs studied (S4 Fig).

**Table 2. Drug exposure before, during and after pregnancy (thousands).**

| | Number of pregnancies, thousands (%) | | | | | | | |
|---|---|---|---|---|---|---|---|---|
| | **Trim-2** | **Trim-1** | **Trim1** | **Trim2** | **Trim3** | **Trim123** | **Trim+1** | **Trim+2** |
| **Number of pregnancies** | 6,015 | 6,015 | 6,015 | 6,015 | 5,985 | 6,015 | 6,015 | 6,015 |
| **Antacids** | 448 (7.4) | 420 (7.0) | 711 (11.8) | 1,022 (17.0) | 1,400 (23.4) | 2,228 (37.0) | 469 (7.8) | 363 (6.0) |
| Locally-acting | 115 (1.9) | 104 (1.7) | 449 (7.5) | 674 (11.2) | 808 (13.5) | 1,541 (25.6) | 107 (1.8) | 97 (1.6) |
| Histamine 2 blocker | 11 (0.2) | 10 (0.2) | 21 (0.3) | 45 (0.7) | 78 (1.3) | 118 (2.0) | 7 (0.1) | 7 (0.1) |
| Proton pump inhibitor | 379 (6.3) | 360 (6.0) | 371 (6.2) | 465 (7.7) | 753 (12.6) | 1,188 (19.7) | 396 (6.6) | 303 (5.0) |
| **Antispasmodics** | 644 (10.7) | 635 (10.6) | 1,387 (23.1) | 1,513 (25.2) | 1,434 (24.0) | 3,097 (51.5) | 1,355 (22.5) | 514 (8.5) |
| Mebeverine | 10 (0.2) | 10 (0.2) | 4 (0.1) | 1 (0.0) | 1 (0.0) | 6 (0.1) | 4 (0.1) | 5 (0.1) |
| Trimebutine | 85 (1.4) | 84 (1.4) | 45 (0.8) | 21 (0.3) | 9 (0.2) | 72 (1.2) | 73 (1.2) | 63 (1.0) |
| Pinaverium | 12 (0.2) | 12 (0.2) | 4 (0.1) | 1 (0.0) | 0 (0.0) | 5 (0.1) | 4 (0.1) | 7 (0.1) |
| Phloroglucinol | 534 (8.9) | 527 (8.8) | 1,346 (22.4) | 1,499 (24.9) | 1,427 (23.8) | 3,065 (51.0) | 1,293 (21.5) | 435 (7.2) |
| Alverine | 57 (1.0) | 58 (1.0) | 26 (0.4) | 7 (0.1) | 3 (0.0) | 34 (0.6) | 22 (0.4) | 37 (0.6) |
| Others | 11 (0.2) | 10 (0.2) | 8 (0.1) | 3 (0.1) | 1 (0.0) | 12 (0.2) | 4 (0.1) | 6 (0.1) |
| **Antinauseants** | 321 (5.3) | 298 (5.0) | 1,548 (25.7) | 385 (6.4) | 191 (3.2) | 1,862 (31.0) | 156 (2.6) | 249 (4.1) |
| Metoclopramide | 50 (0.8) | 42 (0.7) | 805 (13.4) | 170 (2.8) | 83 (1.4) | 971 (16.1) | 22 (0.4) | 35 (0.6) |
| Domperidone | 162 (2.7) | 149 (2.5) | 511 (8.5) | 114 (1.9) | 56 (0.9) | 634 (10.5) | 84 (1.4) | 110 (1.8) |
| Metopimazine | 122 (2.0) | 117 (2.0) | 492 (8.2) | 118 (2.0) | 58 (1.0) | 622 (10.3) | 54 (0.9) | 111 (1.8) |
| 5-HT3 antagonists | 0 (0.0) | 0 (0.0) | 7 (0.1) | 3 (0.0) | 1 (0.0) | 8 (0.1) | 1 (0.0) | 1 (0.0) |
| Others | 0 (0.0) | 0 (0.0) | 0 (0.0) | 0 (0.0) | 0 (0.0) | 0 (0.0) | 1 (0.0) | 1 (0.0) |
| **Laxatives** | 141 (2.4) | 132 (2.2) | 370 (6.2) | 381 (6.3) | 353 (5.9) | 919 (15.3) | 438 (7.3) | 127 (2.1) |
| Lubricant | 8 (0.1) | 8 (0.1) | 13 (0.2) | 8 (0.1) | 5 (0.1) | 25 (0.4) | 18 (0.3) | 9 (0.1) |
| Bulk | 19 (0.3) | 18 (0.3) | 35 (0.6) | 31 (0.5) | 20 (0.3) | 76 (1.3) | 28 (0.5) | 16 (0.3) |
| Osmotic | 108 (1.8) | 101 (1.7) | 315 (5.2) | 325 (5.4) | 234 (3.9) | 741 (12.3) | 356 (5.9) | 98 (1.6) |
| Enema | 30 (0.5) | 27 (0.5) | 59 (1.0) | 59 (1.0) | 126 (2.1) | 226 (3.8) | 93 (1.5) | 23 (0.4) |
| Others | 0 (0.0) | 0 (0.0) | 0 (0.0) | 0 (0.0) | 0 (0.0) | 0 (0.0) | 0 (0.0) | 0 (0.0) |
| **Antidiarrheals** | 275 (4.6) | 263 (4.4) | 287 (4.8) | 230 (3.8) | 141 (2.4) | 618 (10.3) | 158 (2.6) | 244 (4.1) |
| Loperamide | 98 (1.6) | 93 (1.5) | 72 (1.2) | 52 (0.9) | 32 (0.5) | 151 (2.5) | 47 (0.8) | 76 (1.3) |
| Racecadotril | 110 (1.8) | 106 (1.8) | 45 (0.7) | 20 (0.3) | 10 (0.2) | 74 (1.2) | 54 (0.9) | 101 (1.7) |
| Diosmectite | 120 (2.0) | 115 (1.9) | 205 (3.4) | 180 (3.0) | 112 (1.9) | 473 (7.9) | 85 (1.4) | 113 (1.9) |
| **Ursodeoxycholic acid** | 2 (0.0) | 2 (0.0) | 1 (0.0) | 2 (0.0) | 21 (0.3) | 22 (0.4) | 2 (0.0) | 2 (0.0) |
| **All gastrointestinal drugs** | 1,197 (19.9) | 1,147 (19.1) | 2,625 (43.6) | 2,452 (40.8) | 2,603 (43.5) | 4,453 (74.0) | 2,003 (33.3) | 1,006 (16.7) |
| **ATC A02-A09** | 1,208 (20.1) | 1,158 (19.3) | 2,630 (43.7) | 2,457 (40.8) | 2,608 (43.6) | 4,456 (74.1) | 2,018 (33.5) | 1,018 (16.9) |

Medication dispensing before, during and after pregnancy was described for pregnancies ending between April 2010 and June 2018: crude numbers and percentages of exposed pregnancies by trimester. As other drugs for functional gastrointestinal disorders represented <0.1% for each trimester, they were not reported. ATC: Anatomical Therapeutic Chemical.

## Drug exposure over time

The percentage of women who received at least one gastrointestinal drug during pregnancy varied little over time (75.2% in 2010 to 72.0% in 2018). However, exposure to several drugs changed over the study period (Fig 1). Antacid use increased from 34.0% in 2010 to 39.0% in 2018 mainly due to increased use of proton pump inhibitors from 12.2% in 2010 to 26.0% in 2018, while the use of locally-acting antacids and histamine 2 blockers decreased slightly from 27.1% and 2.4% in 2010 to 24.0% and 1.3% in 2018, respectively (Fig 2). Antinauseant drug use decreased from 37.8% in 2010 to 25.2% in 2018, mainly due to decreased use of domperidone (18.3% in 2010 to 2.2% in 2018), while 5-HT3 antagonists use increased from 0.02% in 2010 to 0.4% in 2018 (S5 Fig). Antispasmodic and laxative exposure did not change from 2010 to 2018

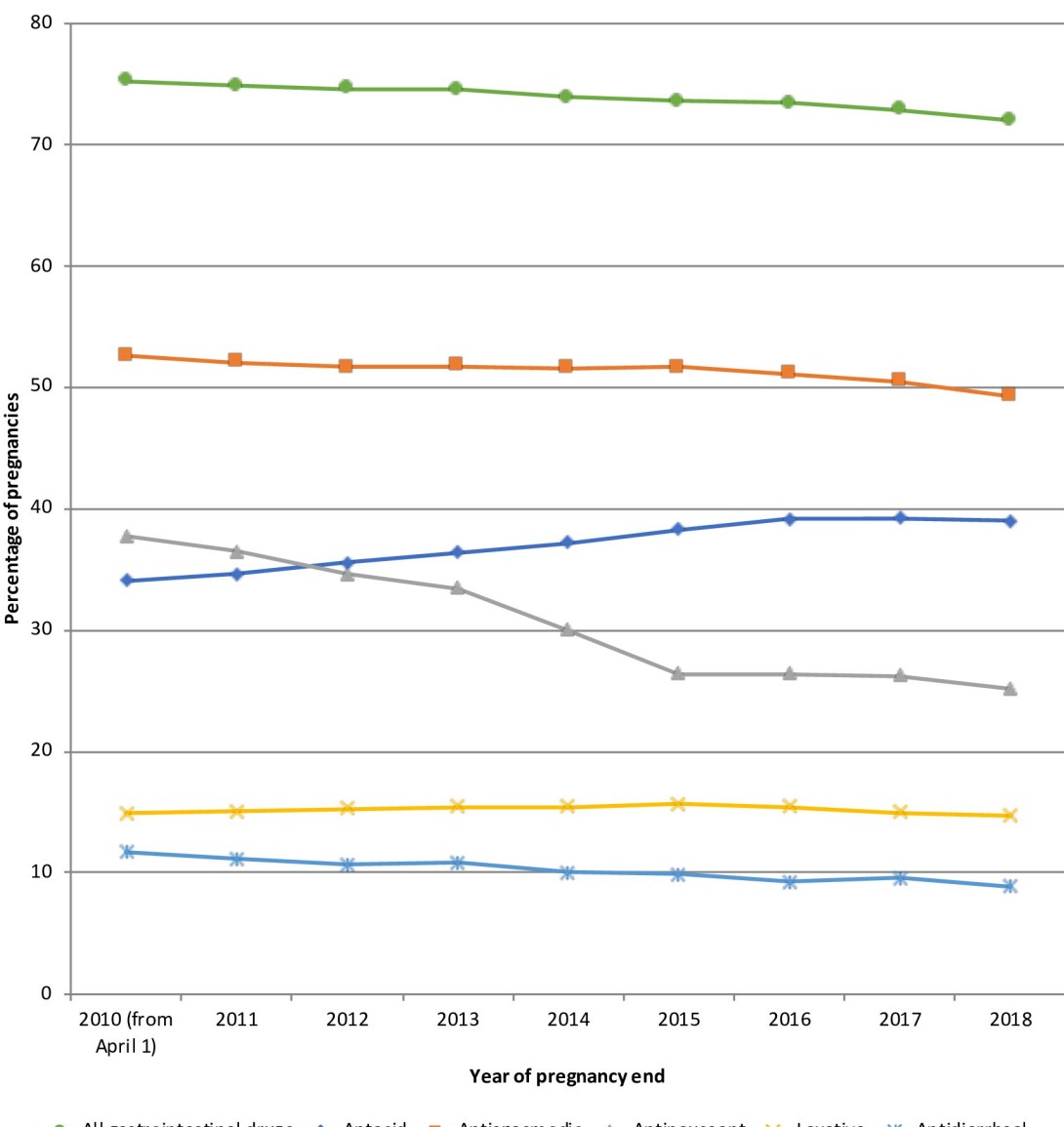

**Fig 1. Drug exposure during pregnancy over time.**

(S6 and S7 Figs). Antidiarrheal drug use decreased slightly from 11.7% in 2010 to 8.9% in 2018, mainly due to decreased use of diosmectite (9.3% in 2010 to 6.4% in 2018) (S8 Fig).

## Drug exposure before, during and after pregnancy

All drug exposures varied during the course of the pregnancy. Antacid use increased from 7.0% during the trimester before pregnancy to 11.8% during the 1st trimester, 17.0% during the 2nd trimester and 23.4% during the 3rd trimester. Antispasmodic use increased from 10.6% during the trimester before pregnancy to 23.1% during the 1st trimester, 25.2% during the 2nd trimester and 24.0% during the 3rd trimester. Antinauseant use increased from 5.0% during the trimester before pregnancy to 25.7% during the 1st trimester, and then decreased to 6.4% during the 2nd trimester and 3.2% during the 3rd trimester. Laxative use increased from 2.2% during the trimester before pregnancy to 6.2% during the 1st trimester, 6.3% during the 2nd

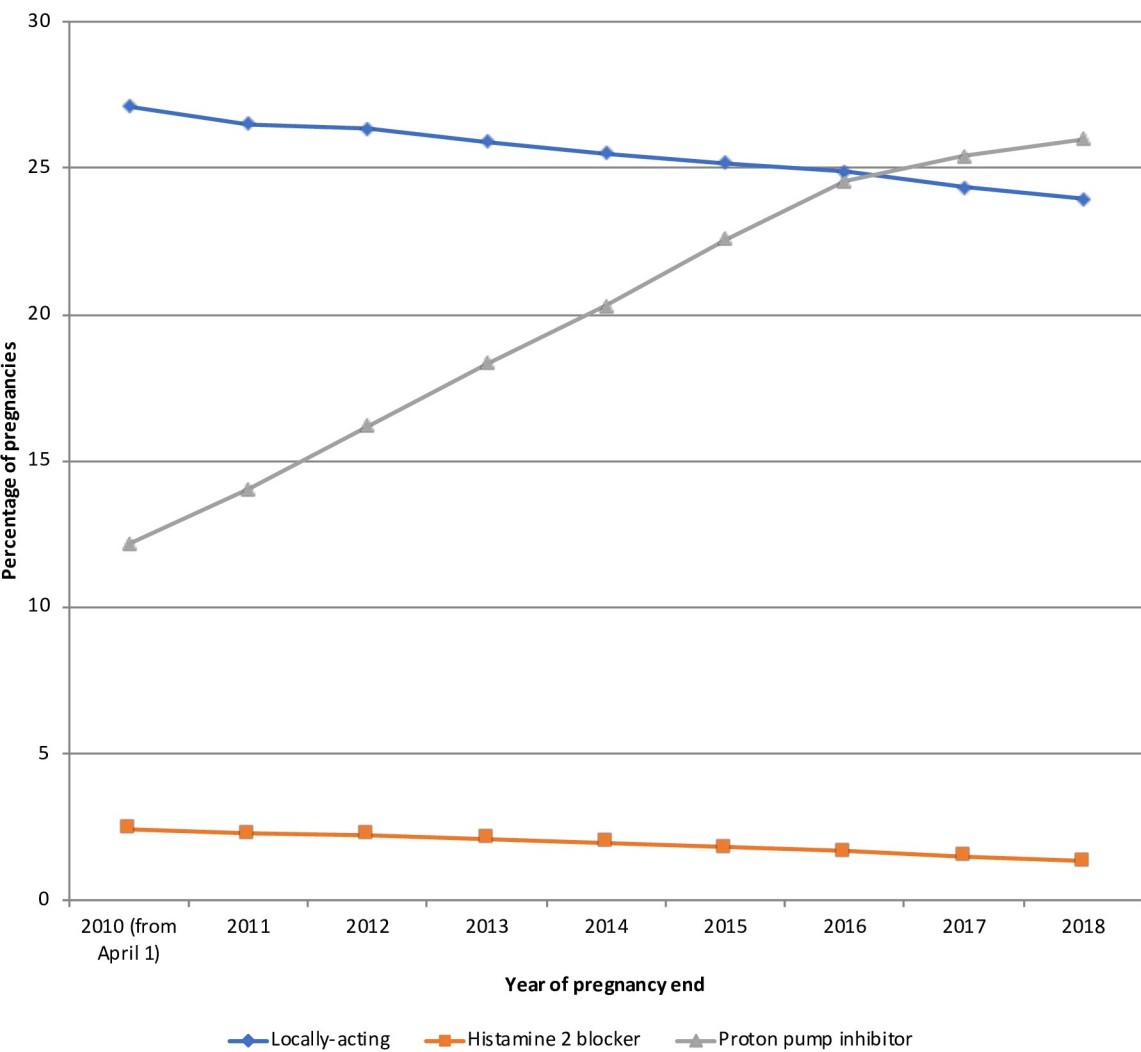

**Fig 2. Antacid exposure during pregnancy over time.**

trimester and 5.9% during the 3rd trimester (Table 2 and Fig 3). These variations in exposure during the course of pregnancy were consistent over age at the beginning of pregnancy, deprivation index and excluding pregnancies lasting less than 37 weeks (S2 and S3 Figs and S4 Table).

## Hospitalizations

Nausea/vomiting was the most common cause of hospitalization for gastrointestinal symptoms or diseases during pregnancy, although it accounted for only 1.0% of pregnancies, mainly during the 1st trimester (0.7%). Hospitalization for cholestasis occurred in 0.3% of pregnancies, mainly in the 3rd trimester (0.3%). The other causes of hospitalization each represented less than 0.2% of pregnancies (Table 3).

## Discussion

This population-based, nationwide study of 6,365,471 pregnancies in France from 2010 to 2018 shows that, during pregnancy, approximately three-quarters of women use drugs for

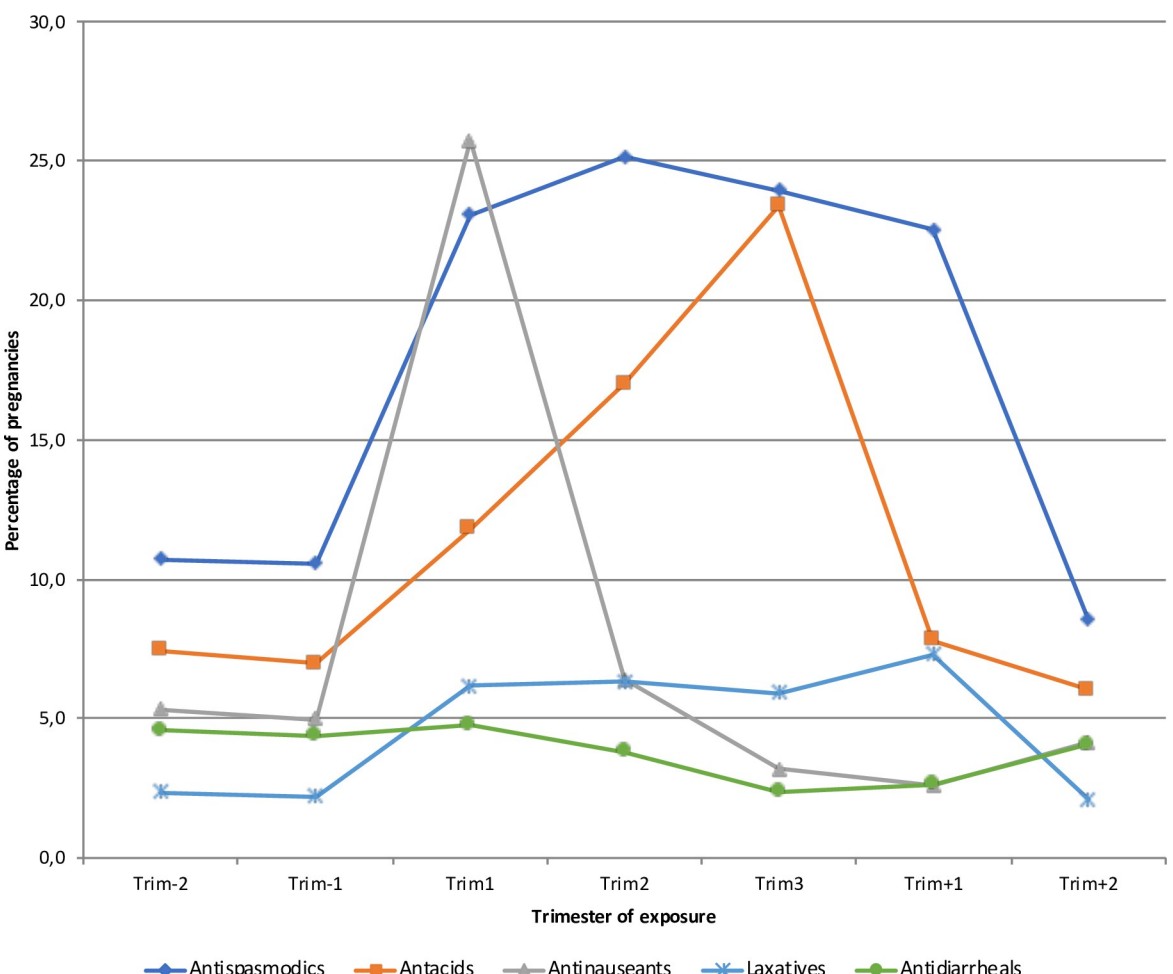

**Fig 3. Drug dispensing before, during and after pregnancy.** Trim: trimester.

gastrointestinal symptoms: approximately one-half of pregnancies are exposed to antispasmodics, and one-third are exposed to antacids and antinauseants. From 2010 to 2018, the use of proton pump inhibitors doubled, while the use of histamine 2 blockers decreased. Prescription of antacids increased during the course of pregnancy, while antinauseants were mainly

**Table 3. Hospitalizations before, during and after pregnancy (thousands).**

| | Number of pregnancies, thousands (%) | | | | | | | |
|---|---|---|---|---|---|---|---|---|
| | Trim-2 | Trim-1 | Trim1 | Trim2 | Trim3 | Trim123 | Trim+1 | Trim+2 |
| **Number of pregnancies** | 6 015 | 6 015 | 6 015 | 6 015 | 5 985 | 6 015 | 6 015 | 6 015 |
| **Nausea/vomiting** | 1 (0.0) | 0 (0.0) | 45 (0.7) | 10 (0.2) | 9 (0.2) | 60 (1.0) | 0 (0.0) | 0 (0.0) |
| **Cholestasis** | 0 (0.0) | 0 (0.0) | 0 (0.0) | 1 (0.0) | 18 (0.3) | 19 (0.3) | 10 (0.2) | 0 (0.0) |
| **Proctologic disease** | 3 (0.1) | 3 (0.0) | 1 (0.0) | 1 (0.0) | 7 (0.1) | 9 (0.2) | 8 (0.1) | 4 (0.1) |
| **Appendicitis** | 3 (0.0) | 2 (0.0) | 2 (0.0) | 1 (0.0) | 0 (0.0) | 3 (0.0) | 3 (0.0) | 2 (0.0) |
| **Biliary tract disease** | 6 (0.1) | 5 (0.1) | 2 (0.0) | 2 (0.0) | 1 (0.0) | 5 (0.1) | 14 (0.2) | 14 (0.2) |

Hospitalizations before, during and after pregnancy was described for pregnancies ending between April 2010 and June 2018: crude numbers and percentages of exposed pregnancies by trimester.

prescribed during the first trimester. Hospitalizations for gastrointestinal diseases were rare, most commonly because of vomiting in 1.0% of pregnancies.

We found that approximately three-quarters of women used drugs for gastrointestinal symptoms during pregnancy. This rate is higher than those reported elsewhere: 4 to 13% in Italy [5, 11], 6% in Ireland [21], 7% in British Columbia [10], 9% in Sweden [7], 8 to 34% in the United States of America [6, 22] and 44% in a worldwide web-based study [23]. Gastrointestinal drug use was already high in 1996 in France, as 69% of women took at least one drug during pregnancy [24, 25]. However, the comparison between countries might be biased by differences in terms of recording of the use, availability and reimbursement of drugs, as well as treatment traditions [26]. Gastrointestinal drugs are widely used in France apart from the context of pregnancy. In 2016, 45% of patients between the ages of 18 and 34 years used at least one gastrointestinal drug, 19% used at least one antacid, 16% used antispasmodics, 7% used antinauseants, 3% used laxatives and 8% used antidiarrheal drugs [27]. These drugs were therefore used more frequently during pregnancy. In addition, the type of drug used during pregnancy is not the same around the world; for example, in some countries, H1 anti-histamines are used as a first line of treatment for nausea while this drug class is not used in France for this indication [28]. We found that drug use was only 5% higher in more deprived pregnancies than in less deprived pregnancies (76.0% vs 72.3% for at least one gastrointestinal drug use); other studies have reported a similar trend, but with more marked differences (around 30%) [23, 29, 30].

Domperidone use decreased from 18.3% in 2010 to 2.2% in 2018, which can be explained by the publication of two studies in 2010 that showed that domperidone prolongs the QT interval, and may cause life-threatening arrhythmias [31, 32]. Metopimazine use decreased in 2015 due to a drug company stock shortage [33].

As pre-marketing clinical trials exclude pregnant women, observational studies are therefore essential to study the efficacy and safety drugs in pregnant women and their offspring. Prescription of proton pump inhibitors doubled between 2010 and 2018 in pregnant women. This increase is of the same order of magnitude as that observed in the general population [34–36]. The efficacy of proton pump inhibitors for the relief of heartburn during pregnancy has not been demonstrated [37] and these drugs should be reserved to women not relieved by lifestyle modification [9, 38]. Several studies [9, 39–41] on the safety of use of proton pump inhibitors during pregnancy are reassuring. However, a recent meta-analysis including 26 observational studies suggested that these drugs were associated with an increased risk of congenital malformations (OR 1.28; 95%CI: 1.09–1.52) [42]. Moreover, toxic effects on animal embryos and fetuses are observed with high doses of omeprazole [43]. The two drugs most commonly used for nausea during pregnancy are dopamine antagonists: metoclopramide and metopimazine. Although metoclopramide has been shown to be effective and safe during pregnancy [9, 28, 44], this is not the case for metopimazine. French recommendations indicate that the use of metopimazine should only be considered after failure of other validated treatments, including metoclopramide and 5-HT3 antagonists [45]. Altogether, these data suggest that a more detailed risk-benefit assessment of prescription during pregnancy is necessary, particularly for proton pump inhibitors and metopimazine which are frequently used during pregnancy.

This study has certain limitations. First, like previously published studies based on the SNDS databases, algorithms rather than clinical data were used to identify pregnancies [2, 16]. Nevertheless, the validity of endpoints such as pregnancy outcomes, maternal age, type of delivery, pregnancy duration and birth weight has previously been demonstrated [16, 19, 46]. Second, over-the-counter drugs or drugs prescribed but not reimbursed (*e.g.* doxylamine) are not included in the SNDS, leading to underestimation of already high rates. Third, a dispensed

drug does not mean that it has been used. However, studying dispensed drugs and not prescriptions avoids primary non-compliance, i.e., that the patient does not redeem the prescription.

The present study also has a number of major strengths. First, it is based on a large and unselected cohort of women with a pregnancy during recent years. Second, the SNDS is a comprehensive database for drug dispensing, covering more than 99% of the French population (around 66,000,000 people).

In conclusion, this large-scale population-based study shows that approximately three-quarters of women are exposed to drugs for gastrointestinal symptoms. From 2010 to 2018, the use of proton pump inhibitors doubled. The benefits and risks associated with increased exposure to proton pump inhibitors during pregnancy need to be further investigated.

## Supporting information

**S1 Fig. Flow-chart.**
(TIF)

**S2 Fig. Number of pregnancies over time.**
(TIF)

**S3 Fig. Drug exposure before, during and after pregnancy by the mother's age at the beginning of pregnancy.** A: 15–24 years; B: 25–34 years; C: 35-49years. Trim: trimester.
(TIF)

**S4 Fig. Drug exposure before, during and after pregnancy by deprivation index.** A: quintile 1 (less deprived); B: quintile 2–4; C: quintile 5 (more deprived). The deprivation index expressed in quintiles was developed in France as the first component of a principal component analysis of 4 socioeconomic variables. Trim: trimester.
(TIF)

**S5 Fig. Ant nauseant exposure during pregnancy over time.**
(TIF)

**S6 Fig. Antispasmodic exposure during pregnancy over time.**
(TIF)

**S7 Fig. Laxative exposure during pregnancy over time.**
(TIF)

**S8 Fig. Antidiarrheal exposure during pregnancy over time.**
(TIF)

**S1 Table. Pregnancy identification algorithms.**
(DOCX)

**S2 Table. Drugs.**
(DOCX)

**S3 Table. Hospitalization for gastrointestinal diseases.**
(DOCX)

**S4 Table. Drug exposure (thousands): Sensitivity analysis excluding pregnancies lasting less than 37 weeks.**
(DOCX)

## Acknowledgments

The authors would like to thank Anthony Saul, MD, for assistance with English grammar and spelling.

## Author Contributions

**Conceptualization:** Antoine Meyer, Alain Weill.

**Data curation:** Jérôme Drouin.

**Formal analysis:** Antoine Meyer.

**Methodology:** Antoine Meyer, Jérôme Drouin, Alain Weill.

**Supervision:** Franck Carbonnel, Alain Weill.

**Validation:** Antoine Meyer, Marion Fermaut, Franck Carbonnel, Alain Weill.

**Writing – original draft:** Antoine Meyer.

**Writing – review & editing:** Antoine Meyer, Marion Fermaut, Franck Carbonnel, Alain Weill.

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
