## [Decision Letter · Decision Letter 0]

1 Dec 2020

PONE-D-20-33435

Drug use for gastrointestinal symptoms during pregnancy: a French nationwide study 2010–2018

PLOS ONE

Dear Dr. Meyer,

Thank you for submitting your manuscript to PLOS ONE. After careful consideration, we feel that it has merit but does not fully meet PLOS ONE’s publication criteria as it currently stands. Therefore, we invite you to submit a revised version of the manuscript that addresses the points raised during the review process.

We look forward to receiving your revised manuscript.

Kind regards,

Angela Lupattelli, PhD

Academic Editor

PLOS ONE

'Conflict of Interest:

-    Antoine Meyer: no conflict of interest.

-    Marion Fermaut: no conflict of interest.

-    Jérôme Drouin: no conflict of interest.

-    Franck Carbonnel: received honoraria from Amgen, BMS, Celltrion, Enterome, Ferring, Janssen, Medtronic, Pfizer, Pharmacosmos, Roche and Tillotts as well as lecture fees from Abbvie, Astra, BMS, Ferring, Janssen, MSD, Pfizer, Pileje, and Takeda.

-    Alain Weill: no conflict of interest.'

a. Please confirm that this does not alter your adherence to all PLOS ONE policies on sharing data and materials, by including the following statement: "This does not alter our adherence to  PLOS ONE policies on sharing data and materials.” (as detailed online in our guide for authors http://journals.plos.org/plosone/s/competing-interests).  If there are restrictions on sharing of data and/or materials, please state these.

Please note that we cannot proceed with consideration of your article until this information has been declared.

Reviewers' comments:

Reviewer's Responses to Questions

**Comments to the Author**

1. Is the manuscript technically sound, and do the data support the conclusions?

Reviewer #1: Yes

Reviewer #2: Yes

Reviewer #3: Yes

2. Has the statistical analysis been performed appropriately and rigorously? 

Reviewer #1: Yes

Reviewer #2: Yes

Reviewer #3: Yes

3. Have the authors made all data underlying the findings in their manuscript fully available?

Reviewer #1: No

Reviewer #2: No

Reviewer #3: No

4. Is the manuscript presented in an intelligible fashion and written in standard English?

Reviewer #1: Yes

Reviewer #2: Yes

Reviewer #3: Yes

5. Review Comments to the Author

Reviewer #1: This is a well-written and interesting paper mapping the dispensation of gastrointestinal drugs during pregnancy in France. It provides useful data as to the prevalence of the use of such drugs and it appears that some of the drugs in particular proton pump inhibitors have increased substantially in recent years. What is missing for me is more justification of why this is important, i.e. what are the potential risks/benefits of understanding patterns in gastrointestinal drug use? It is covered a little bit at the end and it may not be the focus of the paper but the paper needs to justify why understanding patterns of use are important in order to justify why this paper is important.

Specifically:

1. The introduction is very brief and this is where a couple of paragraphs explaining why understanding gastrointestinal drug use is important (ie potential risks/benefits/unknowns) could be added. It would just set up the value of the paper from the start as while a prevalence paper is useful, it is not clear why the issue needs examination at the moment.

Methods, graphs, data all clearly presented and easy to understand. Discussion briefly covers the impact of gastrointestinal drug use in pregnancy. Could also do with some more analysis on why rates are seemingly higher in France and what that could mean in terms of potential risks to mother/child.

Reviewer #2: This study aimed to assess the drug utilization pattern of drugs used to treat gastrointestinal symptoms during pregnancy in France in the period from April 1, 2010 to 2018. As pregnant women are not included in randomized controlled trials for ethical reasons, observational studies are important for assessing drug use and safety of drugs during pregnancy. The data were retrieved from the French national health database, SNDS, that covers more than 99% of the French population. The study population included all pregnancies ending in a live birth or stillbirth after 22 weeks of amenorrhea in the whole study period and thus represents an unselected pregnant population. The study shows that a large proportion of pregnant women in France redeemed prescriptions of drugs used to treat gastrointestinal symptoms in the study period. This is important and clinically relevant information.

Data are not available because access is restricted to French public institutions.

The manuscript is well-written and the methods are sound. However, some revisions and clarifications are required.

Major revisions

1. Hospitalizations for gastrointestinal symptoms or diseases during pregnancy is presented as one of the main results in the manuscript (Table 3), but not mentioned in the abstract. Suggest adding this result to the abstract.

2. In the Materials and Methods section (p3), please add some more information about what information is available regarding pregnancy and birth in the SNDS data source.

3. In the Materials and Methods section (p4), please specify how the trimesters were defined, e.g. in number of days/weeks/months from start of pregnancy. Since trimesters could be defined differently in different studies this is important information to be able to compare studies. Some examples are -90 days to +97 days, 0-90 days, or 0-97 days for the first trimester etc. In the figures and the tables you present additional exposure periods such as Trim-2, Trim-1, Trim 1-3, Trim+1 and Trim+2. Please also clarify the definition of these periods in the Materials and Methods section.

4. In the Materials and Methods section (p6), please clarify how prevalence of use was calculated. What was the denominator? Total number of pregnancies? Add a definition to the statistical analysis paragraph. See also next comment.

5. On page 6, line 139 it is stated that “The unit of analysis was a pregnancy, i.e. all of a patient’s pregnancies were included in the analysis”. Hence, the denominator should be the total number of pregnancies. If this is the denominator, how come the prevalence of use of at least one gastrointestinal drug is reported to be 74% among all pregnancies in the abstract and result section? Shouldn’t it rather be 70% (4,452,779/6,365,471), i.e. the number of pregnancies filling at least one prescription of a gastrointestinal drug divided by the total number of pregnancies included in the study? This goes for the rest of the percentages reported for the specific drug classes as well. Did you use another denominator than the total number of pregnancies for the calculations?

6. Page 6, line 140: What is meant by “inclusion”? All pregnancies in the study period are included and the unit of analysis is a pregnancy. It is my understanding that Table 1 shows baseline characteristics per pregnancy, i.e. a woman could be “included” twice if giving birth twice during the study period. Please clarify.

7. Regarding antinauseants, what is the rational for not including first generation antihistamines such as meclozine (R06AE05) and promethazine (R06AD02)? These drugs are among the most frequently used drugs to treat nausea in pregnancy and they are also considered first line treatment for nausea in pregnancy, at least in some European countries. Not including them will give the wrong picture of antinauseants use in pregnancy (unless these drugs are not used for that purpose in France). Please consider including first generation antihistamines in your study or, if not relevant to the study setting, please add a statement in the manuscript explaining why these drugs were not included.

8. The manuscript could benefit from first presenting a figure showing the overall drug utilization pattern, i.e. the prevalence of use per year for the total use of “all gastrointestinal drugs” and all the drug classes. This figure would be similar to Figure S4, but with the inclusion of the category “all gastrointestinal drugs” as defined in Table S2. Hence, Figure S4 could rather be presented as Figure 1 with the inclusion of “all gastrointestinal drugs”. Then the reader would be presented with the overall picture right away. Figure 1 (flow chart) could rather be moved to supplementary (see minor revisions).

9. Table S2 (and throughout the manuscript): Suggest changing “topical” to “locally-acting” for antacids (A02A, A02BX). Although these antacids are not absorbed systemically, it is not correct to use the term “topical” as they are administered orally and not topically.

10. p12, lines 240-242: In addition to differences in recording drug use and reimbursement of drugs, there could also be differences due to different availability of drugs and treatment traditions. In the Nordic countries for example antispasmodics are rarely used to treat nausea and vomiting in pregnant women, whereas this is the most frequent drug class in France.

11. p14, lines 279-280: What is the scientific evidence for stating that a patient will not use the drug particularly after having read the package leaflet? Do you mean that pregnant women might be more reluctant about taking medications during pregnancy? If so, please rephrase and add a reference after this statement.

12. p14, Suggest to add to the strengths that studying dispensed prescriptions has an advantage compared with studying issued prescriptions as you avoid primary non-compliance, i.e. that the patient does not redeem the prescription.

13. In the Introduction/Discussion section, please add some more information about French recommendations for pharmacological treatment of gastrointestinal symptoms in pregnancy to provide rational for the included drug classes and to discuss whether your results are in accordance with treatment guidelines or recommendations.

14. In general, more explanatory text is needed for figure legends, i.e. more detailed descriptions of what is shown in the figures and explanation of abbreviations etc. Also add labels to x-axis and y-axis, e.g. “Year of birth” for the x-axis in Figure 2.

Minor revisions

1. p3, line 81: Since SNDS contains information on dispensed drugs to outpatients it would be good to add “dispensed” since that would be helpful for the reader to know.

2. p4, line 93: Explain the abbreviation WA.

3. p5, line 121-122 and Table 1: It is a bit confusing that the total number of pregnancies during the study period (“Gravidity”) is presented as 1st, 2nd and 3rd and more pregnancies. This should be 1, 2, 3 or more pregnancies as a woman can only enter one of the categories during the study period, i.e. you refer to the total number of pregnancies and not whether it is the 1st or 2nd pregnancy.

4. p7, line 162: Suggest moving the flow chart (Figure 1) to supplementary. Also, please include “pregnancies” to the textbox at the bottom of the flow chart, i.e. “6,365,471 pregnancies ending with live birth or stillbirth included in the analysis”.

5. p8, line 177: Suggest adding “The antispasmodic phloroglucinol was…” to enhance readability.

6. p9, line 192: What is meant by setron? Does it refer to A04AA? Please use the term from the WHO ATC/DDD index, i.e. “serotonin (5-HT3) antagonists” in the text, tables (Table 2, Table S2, Table S4), and figures (Figure S5) for this drug class.

7. Table 1: suggest changing “Total” (leftmost column) to “Number of pregnancies” so that the reader understands that the unit of measurement is pregnancies and not women. A table should be self-explanatory.

Reviewer #3: This paper describes the consumption of gastrointestinal medication during pregnancy between 2010-2018 in France. Given the high prevalence of medication use during pregnancy and concerns about medication safety, this topic area is very important. Here are a few comments and suggestion to consider

1) The mention of time period is not consistent throughout the paper, during pregnancy, 6 months before to 6 months after pregnancy, trimester -2 to trimester +2. Please clarify

2) Why have the authors chosen to investigate 6 months prior to pregnancy and 6 months postpartum if the objective is medication use during pregnancy. I understand that 3 months prior to pregnancy could be reflective of behavior during the first trimester of pregnancy but there is no justification on to why the authors decided on this particular time period

3) Related to comment 2, how did the authors distinguish between taking the GI medications due to pregnancy-related complications vs. non-pregnancy complications 6 months prior to pregnancy or 6 months postpartum?

4) In line 86 on page 4 please clarify term termination as stated in Table S1

5) In line 109 on page 5 the authors mention the study of the most common reasons for hospitalization for a GI symptom or disease between 6 months before and 6 months after pregnancy. At the bottom of the page in line 133, they mention pregnancy-related hospitalization included hospitalizations during pregnancy for a pregnancy complication. It seems to me that these two variables may have overlaps (for instance, a pregnant woman may get hospitalized due to extreme nausea and vomiting. Please clarify.

6) The statistical analysis section needs to be expanded to explain how medication dispending and hospitalizations during each trimester of pregnancy were described. What method was used? Were the models adjusted for any competing risk factors?

7) Please explain why you decided to excluded pregnancies that did not end in birth. Is there any data to suggest that adverse pregnancy outcomes such as spontaneous abortions, ectopic pregnancy etc occurred as a result of taking GI medications? It would be interesting (as a sensitivity analysis) to determine the % of use of GI medications in pregnancies that were terminated before birth

8) In Line 167, is no income the same as missing? Please clarify in the text and the Table.

9) It is not clear whether “drug exposure” results (lines 174- 178) are presented as a Table or not. If not, it would be more helpful to include a table with these results

10) What methods were used to determine “drug exposure over time”? and are the % in 2010 and 2018 significantly different from each other? Please clarify in the methods section and the results

11) Are the results related to dug exposure before, during, and after pregnancy calculated separately or longitudinally? And are they adjusted or unadjusted %s? Please clarify in methods and the results sections

12) For Table 2, please Add in the footnote of the table how the % were calculated and whether they were adjusted for any of the maternal factors.

13) Table 3, what is N?

6. PLOS authors have the option to publish the peer review history of their article (what does this mean?). If published, this will include your full peer review and any attached files.

Reviewer #1: No

Reviewer #2: **Yes: **Ingvild Odsbu

Reviewer #3: No

---

## [Author Response · Author response to Decision Letter 0]

6 Jan 2021

Comments for Manuscript "Drug use for gastrointestinal symptoms during pregnancy: a French nationwide study 2010–2018"

We thank the academic editor for her comments. We provide hereafter a point-by-point response to these comments, which we hope will clarify and improve the manuscript.

We have adapted our manuscript to meet PLOS ONE's style requirements.

We added the following data availability statement in the cover letter and the manuscript: The authors had access to the SNDS database in application of the provisions of Articles R. 1461-12 et seq. of the French Public Health Code and the French data protection authority decision CNIL-2016-316. Future researchers can request access via the Health data hub: (https://documentation-snds.health-data-hub.fr/introduction/03-acces-snds.html#les-acces-sur-projet).

'Conflict of Interest:

- Antoine Meyer: no conflict of interest.

- Marion Fermaut: no conflict of interest.

- Jérôme Drouin: no conflict of interest.

- Franck Carbonnel: received honoraria from Amgen, BMS, Celltrion, Enterome, Ferring, Janssen, Medtronic, Pfizer, Pharmacosmos, Roche and Tillotts as well as lecture fees from Abbvie, Astra, BMS, Ferring, Janssen, MSD, Pfizer, Pileje, and Takeda.

- Alain Weill: no conflict of interest.'

a. Please confirm that this does not alter your adherence to all PLOS ONE policies on sharing data and materials, by including the following statement: "This does not alter our adherence to PLOS ONE policies on sharing data and materials.” (as detailed online in our guide for authors http://journals.plos.org/plosone/s/competing-interests). If there are restrictions on sharing of data and/or materials, please state these.

Please note that we cannot proceed with consideration of your article until this information has been declared.

We added the following statement in the cover letter and the manuscript: "This does not alter our adherence to PLOS ONE policies on sharing data and materials.”

We added the updated Competing Interests statement in our cover letter.

Reviewers' comments:

Reviewer #1: 

This is a well-written and interesting paper mapping the dispensation of gastrointestinal drugs during pregnancy in France. It provides useful data as to the prevalence of the use of such drugs and it appears that some of the drugs in particular proton pump inhibitors have increased substantially in recent years. What is missing for me is more justification of why this is important, i.e. what are the potential risks/benefits of understanding patterns in gastrointestinal drug use? It is covered a little bit at the end and it may not be the focus of the paper but the paper needs to justify why understanding patterns of use are important in order to justify why this paper is important.

We thank the reviewer for his/her comments. We provide hereafter a point-by-point response to these comments, which we hope will clarify and improve the manuscript.

Specifically:

1. The introduction is very brief and this is where a couple of paragraphs explaining why understanding gastrointestinal drug use is important (ie potential risks/benefits/unknowns) could be added. It would just set up the value of the paper from the start as while a prevalence paper is useful, it is not clear why the issue needs examination at the moment.

Thank you for this important comment. We added a paragraph at the beginning of the introduction to better reflect why understanding gastrointestinal drug use is important:

“Thalidomide was used as an antinauseant medication during pregnancy until its teratogenic effect was demonstrated in the 1960s. This scandal led to an increased awareness of the safe use of drugs during pregnancy. Yet, data on the safety of drugs used during pregnancy (for both women and offspring) is often lacking as pre-marketing clinical trials exclude pregnant women. Observational studies are therefore essential to study the efficacy and safety drugs in pregnant women and their offspring. For instance, a French nationwide study on recent data confirmed that valproic acid, a treatment for epilepsy, cause birth defects and delayed cognitive development.”

We also added this sentence before the aim of the study:

“Drug use studies in pregnancy are useful to identify the most frequently used drugs and thereby decide which drug safety studies should be prioritized.”

Methods, graphs, data all clearly presented and easy to understand. Discussion briefly covers the impact of gastrointestinal drug use in pregnancy. Could also do with some more analysis on why rates are seemingly higher in France and what that could mean in terms of potential risks to mother/child.

That’s a very good point and it is addressed in the discussion: “We found that approximately three-quarters of women used drugs for gastrointestinal symptoms during pregnancy. This rate is higher than those reported elsewhere: 4 to 13% in Italy, 6% in Ireland, 7% in British Columbia, 9% in Sweden, 8 to 34% in the United States of America and 44% in a worldwide web-based study. Gastrointestinal drug use was already high in 1996 in France, as 69% of women took at least one drug during pregnancy. However, the comparison between countries might be biased by differences in terms of recording of the use, availability and reimbursement of drugs, as well as treatment traditions.”

We have modified a sentence in the discussion to better reflect the potential risks to mother/child: 

Previous sentence in the discussion: Altogether, these data suggest that a more detailed risk-benefit assessment of prescription during pregnancy is necessary, particularly for proton pump inhibitors and metopimazine.

New sentence in the discussion: Altogether, these data suggest that a more detailed risk-benefit assessment of prescription during pregnancy is necessary, particularly for proton pump inhibitors and metopimazine which are frequently used during pregnancy.

Reviewer #2: 

This study aimed to assess the drug utilization pattern of drugs used to treat gastrointestinal symptoms during pregnancy in France in the period from April 1, 2010 to 2018. As pregnant women are not included in randomized controlled trials for ethical reasons, observational studies are important for assessing drug use and safety of drugs during pregnancy. The data were retrieved from the French national health database, SNDS, that covers more than 99% of the French population. The study population included all pregnancies ending in a live birth or stillbirth after 22 weeks of amenorrhea in the whole study period and thus represents an unselected pregnant population. The study shows that a large proportion of pregnant women in France redeemed prescriptions of drugs used to treat gastrointestinal symptoms in the study period. This is important and clinically relevant information.

Data are not available because access is restricted to French public institutions.

The manuscript is well-written and the methods are sound. However, some revisions and clarifications are required.

We thank the reviewer for her comments. We provide hereafter a point-by-point response to these comments, which we hope will clarify and improve the manuscript.

Major revisions

1. Hospitalizations for gastrointestinal symptoms or diseases during pregnancy is presented as one of the main results in the manuscript (Table 3), but not mentioned in the abstract. Suggest adding this result to the abstract.

We added this sentence in the methods section of the abstract: “We also assessed hospitalization for gastrointestinal symptoms during pregnancy.”

We added this sentence in the results section of the abstract: “Nausea/vomiting was the most common cause of hospitalization for gastrointestinal symptoms or diseases during pregnancy, although it accounted for only 1.0% of pregnancies.”

2. In the Materials and Methods section (p3), please add some more information about what information is available regarding pregnancy and birth in the SNDS data source.

The information available in our database and used in this study regarding pregnancy and birth were included in the methods in the sections: data source, study population and maternal and pregnancy characteristics. We have added references to the content of the database that is not used in our study: Blotiere 2018, Blotiere 2019, Blotiere 2020.

3. In the Materials and Methods section (p4), please specify how the trimesters were defined, e.g. in number of days/weeks/months from start of pregnancy. Since trimesters could be defined differently in different studies this is important information to be able to compare studies. Some examples are -90 days to +97 days, 0-90 days, or 0-97 days for the first trimester etc. In the figures and the tables you present additional exposure periods such as Trim-2, Trim-1, Trim 1-3, Trim+1 and Trim+2. Please also clarify the definition of these periods in the Materials and Methods section.

We added in the methods how trimesters were defined.

Previous sentence in the methods: We studied each drug dispensed for gastrointestinal symptoms in the 6 months before, during and in the six months after pregnancy, for each trimester, including: antacids, antispasmodics, antinauseants, laxatives, antidiarrheals and other drugs for functional gastrointestinal disorders. Details on International Non-Proprietary name and Anatomical Therapeutic Chemical Classification are given in S2 Table.

New sentence in the methods: We studied each drug dispensed for gastrointestinal symptoms including: antacids, antispasmodics, antinauseants, laxatives, antidiarrheals and other drugs for functional gastrointestinal disorders. Details on International Non-Proprietary name and Anatomical Therapeutic Chemical Classification are given in S2 Table. For each pregnancy, seven trimesters were studied: two trimesters before the beginning of pregnancy (Trim-2: day −182 to day −92; Trim-1: day −91 to day −1), each trimester of pregnancy (Trim1: day 0 i.e. fertilisation to day 90; Trim2: day 91 to day 181; Trim3: day 182 to delivery −1) and two trimesters after the end of pregnancy (Trim+1: delivery to delivery + 91; Trim+2: delivery + 92 to delivery + 182).

4. In the Materials and Methods section (p6), please clarify how prevalence of use was calculated. What was the denominator? Total number of pregnancies? Add a definition to the statistical analysis paragraph. See also next comment.

We clarified this definition in the statistical analysis section of the methods.

Previous sentence in the methods: Medication dispensing and hospitalizations during each trimester of pregnancy were then described.

New sentence in the methods: Medication dispensing during pregnancy over time was then described for pregnancy ending between April 2010 and December 2018: crude numbers and percentages of exposed pregnancies by year of pregnancy end. Medication dispensing and hospitalizations before, during and after pregnancy was described for pregnancies ending between April 2010 and June 2018: crude numbers and percentages of exposed pregnancies by trimester.

5. On page 6, line 139 it is stated that “The unit of analysis was a pregnancy, i.e. all of a patient’s pregnancies were included in the analysis”. Hence, the denominator should be the total number of pregnancies. If this is the denominator, how come the prevalence of use of at least one gastrointestinal drug is reported to be 74% among all pregnancies in the abstract and result section? Shouldn’t it rather be 70% (4,452,779/6,365,471), i.e. the number of pregnancies filling at least one prescription of a gastrointestinal drug divided by the total number of pregnancies included in the study? This goes for the rest of the percentages reported for the specific drug classes as well. Did you use another denominator than the total number of pregnancies for the calculations?

The answer to the previous question also answers this question. Medication dispensing before, during and after pregnancy was described for pregnancies ending between April 2010 and June 2018. Indeed, pregnancies ending between July 2018 and December 2018 were excluded to have the full seven trimesters of exposure for each pregnancy. Therefore, 74.0% (4,452,779 / 6,014,811) received at least one gastrointestinal drug during pregnancy as shown in Table 2.

6. Page 6, line 140: What is meant by “inclusion”? All pregnancies in the study period are included and the unit of analysis is a pregnancy. It is my understanding that Table 1 shows baseline characteristics per pregnancy, i.e. a woman could be “included” twice if giving birth twice during the study period. Please clarify.

You understood correctly. We modified a sentence in the methods to further clarify this point:

Previous sentence in the methods: We first described maternal and pregnancy characteristics at inclusion: median and interquartile range (IQR) for continuous variables and proportions for dichotomous and categorical variables.

New sentence in the methods: We first described maternal characteristics at the beginning of pregnancy and pregnancy characteristics for each pregnancy: median and interquartile range (IQR) for continuous variables and proportions for categorical variables.

7. Regarding antinauseants, what is the rational for not including first generation antihistamines such as meclozine (R06AE05) and promethazine (R06AD02)? These drugs are among the most frequently used drugs to treat nausea in pregnancy and they are also considered first line treatment for nausea in pregnancy, at least in some European countries. Not including them will give the wrong picture of antinauseants use in pregnancy (unless these drugs are not used for that purpose in France). Please consider including first generation antihistamines in your study or, if not relevant to the study setting, please add a statement in the manuscript explaining why these drugs were not included.

H1 anti-histamines drugs such as meclozine and promethazine are not used in France for nausea as they are not authorized for use in this indication.

We added a sentence in the discussion to clarify this point: “In addition, the type of drug used during pregnancy is not the same around the world; for example, in some countries, H1 anti-histamines are used as a first line of treatment for nausea while this drug class is not used in France for this indication.”

8. The manuscript could benefit from first presenting a figure showing the overall drug utilization pattern, i.e. the prevalence of use per year for the total use of “all gastrointestinal drugs” and all the drug classes. This figure would be similar to Figure S4, but with the inclusion of the category “all gastrointestinal drugs” as defined in Table S2. Hence, Figure S4 could rather be presented as Figure 1 with the inclusion of “all gastrointestinal drugs”. Then the reader would be presented with the overall picture right away. Figure 1 (flow chart) could rather be moved to supplementary (see minor revisions).

Again, a very good point. We have followed your recommendations by adding an "all gastrointestinal drugs" category to S4 Fig which is now Fig 1. Flow-chart has been moved to supplementary

9. Table S2 (and throughout the manuscript): Suggest changing “topical” to “locally-acting” for antacids (A02A, A02BX). Although these antacids are not absorbed systemically, it is not correct to use the term “topical” as they are administered orally and not topically.

We replaced “topical” to “locally-acting” for antacids.

10. p12, lines 240-242: In addition to differences in recording drug use and reimbursement of drugs, there could also be differences due to different availability of drugs and treatment traditions. In the Nordic countries for example antispasmodics are rarely used to treat nausea and vomiting in pregnant women, whereas this is the most frequent drug class in France.

We added availability of drugs and treatment traditions in the difference between countries in the discussion.

Previous sentence in the discussion: However, the comparison between countries might be biased by differences in terms of recording of the use and reimbursement of drugs.

New sentence in the discussion: However, the comparison between countries might be biased by differences in terms of recording of the use, availability and reimbursement of drugs, as well as treatment traditions.

11. p14, lines 279-280: What is the scientific evidence for stating that a patient will not use the drug particularly after having read the package leaflet? Do you mean that pregnant women might be more reluctant about taking medications during pregnancy? If so, please rephrase and add a reference after this statement.

We have removed the end of the sentence in the discussion concerning the package leaflet, in fact, there can be several reasons for not taking a delivered medication.

Previous sentence in the discussion: Third, a dispensed drug does not mean that it has been used, particularly after the patient has read the package leaflet.

New sentence in the discussion: Third, a dispensed drug does not mean that it has been used. However, studying dispensed drugs and not prescriptions avoids primary non-compliance, i.e., that the patient does not redeem the prescription.

12. p14, Suggest to add to the strengths that studying dispensed prescriptions has an advantage compared with studying issued prescriptions as you avoid primary non-compliance, i.e. that the patient does not redeem the prescription.

We added this strength in the discussion:

Previous sentence in the discussion: Third, a dispensed drug does not mean that it has been used, particularly after the patient has read the package leaflet.

New sentence in the discussion: Third, a dispensed drug does not mean that it has been used. However, studying dispensed drugs and not prescriptions avoids primary non-compliance, i.e., that the patient does not redeem the prescription.

13. In the Introduction/Discussion section, please add some more information about French recommendations for pharmacological treatment of gastrointestinal symptoms in pregnancy to provide rational for the included drug classes and to discuss whether your results are in accordance with treatment guidelines or recommendations.

Unfortunately, there is no French recommendation on the use of gastrointestinal medications during pregnancy.

14. In general, more explanatory text is needed for figure legends, i.e. more detailed descriptions of what is shown in the figures and explanation of abbreviations etc. Also add labels to x-axis and y-axis, e.g. “Year of birth” for the x-axis in Figure 2.

To increase clarity, we added labels to x-axis and y-axis, description and explanation of abbreviations for all figures.

Minor revisions

1. p3, line 81: Since SNDS contains information on dispensed drugs to outpatients it would be good to add “dispensed” since that would be helpful for the reader to know.

We modified this sentence in the methods to increase clarity:

Previous sentence in the methods: The SNDS contains all outpatient information (demographics, drugs, and procedures) and all inpatient information (expensive drugs dispensed, procedures performed during hospital stays, and diagnoses).

New sentence in the methods: The SNDS contains all outpatient information (demographics, drugs dispensed, and procedures) and all inpatient information (expensive drugs dispensed, procedures performed during hospital stays, and diagnoses).

2. p4, line 93: Explain the abbreviation WA.

We replaced all the “WA” by “weeks” throughout the manuscript and specified in the methods that they were weeks of amenorrhea.

Previous sentence in the methods: All pregnancies ending with a birth - either livebirth or stillbirth (delivery of a dead fetus after 22WA) - between April 2010 and December 2018 were included.

New sentence in the methods: All pregnancies ending with a birth - either livebirth or stillbirth - between April 2010 and December 2018 were included. A stillbirth was the delivery of a dead fetus after 22 weeks of amenorrhea (referred to as weeks thereafter).

3. p5, line 121-122 and Table 1: It is a bit confusing that the total number of pregnancies during the study period (“Gravidity”) is presented as 1st, 2nd and 3rd and more pregnancies. This should be 1, 2, 3 or more pregnancies as a woman can only enter one of the categories during the study period, i.e. you refer to the total number of pregnancies and not whether it is the 1st or 2nd pregnancy.

Gravidity refers here as it is the 1st, 2nd,… pregnancy during the study period. We modified a sentence in the methods to increase clarity:

Previous sentence in the methods: Gravidity (1st pregnancy, 2nd and more pregnancy) was defined as the number pregnancies in a woman during the study period.

New sentence in the methods: Gravidity (1st pregnancy, 2nd pregnancy,…) was defined as the pregnancy number in a woman during the study period.

4. p7, line 162: Suggest moving the flow chart (Figure 1) to supplementary. Also, please include “pregnancies” to the textbox at the bottom of the flow chart, i.e. “6,365,471 pregnancies ending with live birth or stillbirth included in the analysis”.

We moved the flow-chart to supplementary and modified the bottom box as suggested: “6,365,471 pregnancies ending with live birth or stillbirth included in the analysis”.

5. p8, line 177: Suggest adding “The antispasmodic phloroglucinol was…” to enhance readability.

We modified this sentence to enhance readability:

Previous sentence in the results: Phloroglucinol was by far the most commonly prescribed drug (51.0% of all pregnancies).

Previous sentence in the results: The antispasmodic phloroglucinol was by far the most commonly prescribed drug (51.0% of all pregnancies).

6. p9, line 192: What is meant by setron? Does it refer to A04AA? Please use the term from the WHO ATC/DDD index, i.e. “serotonin (5-HT3) antagonists” in the text, tables (Table 2, Table S2, Table S4), and figures (Figure S5) for this drug class.

We replaced “setron” by “5-HT3 antagonists” throughout the manuscript.

7. Table 1: suggest changing “Total” (leftmost column) to “Number of pregnancies” so that the reader understands that the unit of measurement is pregnancies and not women. A table should be self-explanatory.

We replaced “Total” by “Number of pregnancies” in Table 1.

Reviewer #3: 

This paper describes the consumption of gastrointestinal medication during pregnancy between 2010-2018 in France. Given the high prevalence of medication use during pregnancy and concerns about medication safety, this topic area is very important. Here are a few comments and suggestion to consider

We thank the reviewer for his/her comments. We provide hereafter a point-by-point response to these comments, which we hope will clarify and improve the manuscript.

1) The mention of time period is not consistent throughout the paper, during pregnancy, 6 months before to 6 months after pregnancy, trimester -2 to trimester +2. Please clarify

We replaced “6 months” by “2 trimesters” throughout the manuscript.

2) Why have the authors chosen to investigate 6 months prior to pregnancy and 6 months postpartum if the objective is medication use during pregnancy. I understand that 3 months prior to pregnancy could be reflective of behavior during the first trimester of pregnancy but there is no justification on to why the authors decided on this particular time period

We studied three periods: before, during and after pregnancy because we wanted not only to study drug exposure during pregnancy but also to compare it with that outside of pregnancy. The choice of two trimesters before and after pregnancy was arbitrary, but it seemed to us that it could capture drug exposure before and during pregnancy.

3) Related to comment 2, how did the authors distinguish between taking the GI medications due to pregnancy-related complications vs. non-pregnancy complications 6 months prior to pregnancy or 6 months postpartum?

In this study we cannot know what was the cause of the gastrointestinal symptom that led to the use of these gastrointestinal drugs, but this was not the aim of our study, we simply studied drug use for gastrointestinal symptoms during pregnancy. Drug use studies in pregnancy are useful to identify the most frequently used drugs and thereby decide which drug safety studies to conduct.

4) In line 86 on page 4 please clarify term termination as stated in Table S1

As there was no pregnancy termination (abortions, ectopic pregnancies,…) included in our study, we simplified this sentence to increase clarity:

Previous sentence in the methods: The pregnancy end date was the date of delivery or termination of pregnancy, or, when missing, the date of admission for pregnancy completion.

New sentence in the methods: The pregnancy end date was the date of delivery, or, when missing, the date of admission for pregnancy completion.

5) In line 109 on page 5 the authors mention the study of the most common reasons for hospitalization for a GI symptom or disease between 6 months before and 6 months after pregnancy. At the bottom of the page in line 133, they mention pregnancy-related hospitalization included hospitalizations during pregnancy for a pregnancy complication. It seems to me that these two variables may have overlaps (for instance, a pregnant woman may get hospitalized due to extreme nausea and vomiting. Please clarify.

That’s a very good point. As these two variables may overlap and as pregnancy-related hospitalizations are not studied here, we removed pregnancy-related hospitalizations in the methods and in Table 1.

6) The statistical analysis section needs to be expanded to explain how medication dispending and hospitalizations during each trimester of pregnancy were described. What method was used? Were the models adjusted for any competing risk factors?

As requested by reviewer 2, we have completed the methods to specify the statistical analyses performed. No adjustments were made because the objective of the study was to study drug use in the entire French population, we added the word “crude” in the statistical analysis section to clarify this point.

We added in the statistical analysis section: “Medication dispensing during pregnancy over time was then described for pregnancy ending between April 2010 and December 2018: crude numbers and percentages of exposed pregnancies by year of pregnancy end. Medication dispensing and hospitalizations before, during and after pregnancy was described for pregnancies ending between April 2010 and June 2018: crude numbers and percentages of exposed pregnancies by trimester.”

7) Please explain why you decided to excluded pregnancies that did not end in birth. Is there any data to suggest that adverse pregnancy outcomes such as spontaneous abortions, ectopic pregnancy etc occurred as a result of taking GI medications? It would be interesting (as a sensitivity analysis) to determine the % of use of GI medications in pregnancies that were terminated before birth

Point well taken. Whether gastroenterological medications cause more abortions or more ectopic pregnancies is an important question, but this was not the aim of our study, which was a descriptive analysis of drug use during pregnancy. We excluded these pregnancies because to study drug exposure throughout pregnancy, pregnancies that last three trimesters are needed. We added this precision in the methods: 

“We excluded pregnancies with elective or therapeutic abortions, spontaneous abortions, ectopic pregnancies and hydatidiform moles and other abnormal products of conception (blighted ovum and nonhydatidiform moles) because to study drug exposure throughout pregnancy, pregnancies that last three trimesters are needed.”

A sensitivity analysis was also performed excluding pregnancies of less than 37 weeks to have only pregnancies with three full trimesters (S4 Table). We added this precision in the statistical analysis section of the methods: “A sensitivity analysis was performed and excluded pregnancies of less than 37 weeks.”

8) In Line 167, is no income the same as missing? Please clarify in the text and the Table.

No income is not the same as missing. No income means that the woman did not receive a salary before her maternity leave. To increase clarity, we modified a sentence in the methods:

Previous sentence in the methods: … income (none, <€2,000/month, ≥€2,000/month or missing value) calculated from the woman's salary during the three months before maternity leave. Income was only available for pregnancies for which women received maternity benefits, i.e., pregnancies among general health insurance scheme beneficiaries (93.7% of pregnancies).

New sentence in the methods: … income (general health insurance scheme: none, <€2,000/month, ≥€2,000/month; and agricultural/self-employed scheme) calculated from the woman's salary during the three months before maternity leave.

We also modified Table 1 accordingly.

9) It is not clear whether “drug exposure” results (lines 174- 178) are presented as a Table or not. If not, it would be more helpful to include a table with these results

These results are included in Table 2. We added the precision in the results to increase clarity.

Previous sentence in the results: Among all pregnancies with delivery, 4,452,779 (74.0%) received at least one gastrointestinal drug during pregnancy; 2,228,275 (37.0%) pregnancies were exposed to an antacid, 3,096,858 (51.5%) to an antispasmodic, 1,861,731 (31.0%) to an antinauseant, 919,116 (15.3%) to a laxative and 617,808 (10.3%) to an antidiarrheal.

New sentence in the results: Among all pregnancies with delivery, 4,452,779 (74.0%) received at least one gastrointestinal drug during pregnancy; 2,228,275 (37.0%) pregnancies were exposed to an antacid, 3,096,858 (51.5%) to an antispasmodic, 1,861,731 (31.0%) to an antinauseant, 919,116 (15.3%) to a laxative and 617,808 (10.3%) to an antidiarrheal (Table 2).

10) What methods were used to determine “drug exposure over time”? and are the % in 2010 and 2018 significantly different from each other? Please clarify in the methods section and the results

We added a sentence in the statistical analysis section of the methods to define drug exposure over time: “Medication dispensing during pregnancy over time was then described for pregnancy ending between April 2010 and December 2018: crude numbers and percentages of exposed pregnancies by year of pregnancy end.”

However, we did not perform statistical testing since our objective was descriptive. In addition, given the very large sample size of the study (6,365,471), clinically irrelevant differences would be statistically significant.

11) Are the results related to dug exposure before, during, and after pregnancy calculated separately or longitudinally? And are they adjusted or unadjusted %s? Please clarify in methods and the results sections

We added a sentence in the statistical analysis section of the methods to define drug exposure before, during, and after pregnancy: “Medication dispensing and hospitalizations before, during and after pregnancy was described for pregnancies ending between April 2010 and June 2018: crude numbers and percentages of exposed pregnancies by trimester.” No adjustments were made because the objective of the study was to study drug use in the entire French population, we added the word “crude” in the statistical analysis section to clarify this point.

12) For Table 2, please Add in the footnote of the table how the % were calculated and whether they were adjusted for any of the maternal factors.

We added a footnote in Table 2, Table 3 and S4 Table to explain how percentages were calculated: “Medication dispensing before, during and after pregnancy was described for pregnancies ending between April 2010 and June 2018: crude numbers and percentages of exposed pregnancies by trimester.”.

13) Table 3, what is N?

We replaced N by Number of pregnancies to increase clarity in Table 1, Table 2, Table 3 and S4 Table.

---

## [Editor Report · Decision Letter 1]

11 Jan 2021

Drug use for gastrointestinal symptoms during pregnancy: a French nationwide study 2010–2018

PONE-D-20-33435R1

Dear Dr. Meyer,

We’re pleased to inform you that your manuscript has been judged scientifically suitable for publication and will be formally accepted for publication once it meets all outstanding technical requirements.

Kind regards,

Angela Lupattelli, PhD

Academic Editor

PLOS ONE

---

## [Editor Report · Acceptance letter]

12 Jan 2021

PONE-D-20-33435R1 

Drug use for gastrointestinal symptoms during pregnancy: a French nationwide study 2010–2018 

Dear Dr. Meyer:

I'm pleased to inform you that your manuscript has been deemed suitable for publication in PLOS ONE. Congratulations! Your manuscript is now with our production department. 

Kind regards, 

on behalf of

Dr. Angela Lupattelli 

Academic Editor

PLOS ONE